# Timelessness Strictly inside the Quantum Realm

**DOI:** 10.3390/e23060772

**Published:** 2021-06-18

**Authors:** Knud Thomsen

**Affiliations:** Paul Scherrer Institute (PSI), 5232 Villigen, Switzerland; knud.thomsen@psi.ch

**Keywords:** relational time, timelessness, records, causality in 1 real world, interpretations, Big Bang, inflation

## Abstract

Time is one of the undisputed foundations of our life in the real world. Here it is argued that inside small isolated quantum systems, time does not pass as we are used to, and it is primarily in this sense that quantum objects enjoy only limited reality. Quantum systems, which we know, are embedded in the everyday classical world. Their preparation as well as their measurement-phases leave durable records and traces in the entropy of the environment. The Landauer Principle then gives a quantitative threshold for irreversibility. With double slit experiments and tunneling as paradigmatic examples, it is proposed that a label of timelessness offers clues for rendering a Copenhagen-type interpretation of quantum physics more “realistic” and acceptable by providing a coarse but viable link from the fundamental quantum realm to the classical world which humans directly experience.

There are at least two levels of uncertainty associated with Quantum Physics since its discovery, and they are not independent of each other. The basic quantum formalisms are well-established and they comprise intrinsic quantified uncertainty relations. The observed behavior of quantum systems seems weird and runs in many cases completely counter to expectations directly based on everyday experience. Interpretations of quantum mechanics try to bridge that gulf. Interpretations of the formalism are many and partly mutually exclusive or even contradictory [1]. From this it ensues that on an effective meta level, at the “common-sense” end of the scale, disorientation and uncertainty prevail. The “meaning” of quantum physics is unclear, a comprehensive embedding in the form of an understandable relation to human everyday conceptions seems beyond reach. Still, without touching the formalism, such shall be attempted to sketch The aim is to draw a crude but overarching picture, allowing us to relate the world, which humans now bodily inhabit and which is aptly described by classical physics, with its seemingly bizarre foundations in quantum mechanics [2,3].

Keeping to the overwhelming naive observational evidence and leaving Einstein relativity aside for a start, one can coarsely outline a distinction between two major domains.

## 1. Classical World, CM

Massive bodies occupy single well-defined positions; there are macroscopically distinguishable states, quantities like mass, energy and momentum can assume continuous values, thermodynamics provides the foundation with global irreversibility and an undeniable arrow of time according to the Second Law of Thermodynamics, entailing causal order; measurements do not perturb the measured, and Special and General Relativity deliver the best models of the spacetime-background with Newtonian space and time as almost perfect approximations for daily use.

## 2. Quantum Realm, QM

Many quantities, like energy and spin, come in discrete packets—they are quantized. In some sense, mostly microscopic systems are rather fragile and undisturbed (most often isolated) ones are described by the linear Schrödinger Equation, characterizing a unitary evolution of the wavefunction, superpositions of wavefunctions, entanglement, decoherence, reversibility, non-locality and uncertainty relations for joint measurements of non-commuting observables. Measuring entails back-action to a measured system, and the Born rule applies when obtaining definitive results in the form of individual random single outcomes of measurements.

It goes without saying that this distinction is referring to theories and descriptions and their respective scopes of application and not to Kant’s “Ding an sich”. Even when starting with a rather broad notion of classical (and quantum) phenomena, some intrinsic quantum mechanical aspects, which cannot be understood in classical terms, remain [4]. It is the interface/borderline between these sectors which is most interesting. Therein also lies the core of the measurement problem.

The Copenhagen interpretation postulates a disruptive collapse of the wavefunction, whereas decoherence accounts, as the seemingly major alternative, formally stay inside quantum mechanics and expound why a result is effectively (for all practical purposes) as good as a genuine classical state [5].

Decoherence does not entirely resolve the measurement problem as the mechanism by itself does not explain the occurrence of definite outcomes according to the Born rule; the composite system remains a superposition, and at least some robust entangled states are still reversible in principle [6,7,8,9]. This latter point can be remedied by considering that there are limits to arbitrary “entanglement dilution”. At some stage there are so many objects involved that states become too numerous to handle for reversal in a finite universe. For effectively indistinguishable states, there is no way of reversing a development, although entanglement can help distinguish orthogonal product states [10,11,12]. The Montevideo interpretation, for example, suggests something like this by assuming an impact of quantum gravity, which results in fundamental limitations for the accuracy of clocks [13]. Already some time ago, Roger Penrose has estimated finite life times of the superposition states of masses effected by gravity [14].

Different thresholds from criteria for entanglement can be given, while limits to easily detect existing entanglement in the presence of noise have been pointed out recently [15,16,17].

Mixing these levels of QM and CM, and stepping back and forth between the relevant descriptions can easily lead to inconsistencies and contradictions. A recent thought experiment has highlighted that the three naively innocent assumptions of Universality, Consistency and Uniqueness apparently cannot be met simultaneously by plain unitary quantum physics applied indiscriminately to itself [18].

In a short paper, it has been argued that emphasizing complementarity and keeping with a fundamental distinction between QM and CM, accepting the importance of a clear Heisenberg cut, one can avoid the purported contradictions [19]. The conclusion offered was that, indeed, quantum mechanics cannot serve as the best description for all of reality, and that the described thought experiment features a truly shifting split; it does not implement one overall consistent setting or application of the admissible rules, in particular with respect to time [20,21,22,23,24].

When a measurement has been performed and the outcome is determined (and remembered), an outside observer not knowing that outcome, can assign classical probabilities to it, but cannot put the total system - including an inside classical observer—in superposition, with humans living in CM. A version of the experiment, where these are replaced by fully reversible quantum computers and thus remaining completely inside QM, would be principally different [11,12,19,25].

This line of argument redeems Schrödinger’s cat from its unpleasant status: the constellation would just describe an improper use of the quantum formalism when including a truly macroscopic entity, where a lot of entropy is generated and time-reversal of the full system simply is not possible. Isolated is only the decaying nucleus; the relevant observer and a record is established by the poor animal. The uncertainty for an external observer is purely epistemological, only secondary to the one of the quantum system hidden inside the sealed arrangement.

“We just cannot have classical and quantum behavior at the same TIME” was proposed as catchphrase for epitomizing “timelessness” inside QM. The argument is that, with time starting anew with each collapse at the strongly asymmetric transition from QM → CM, the very concept of time inside QM appears questionable [19]. At the same time, some clear transient from QM → CM makes any tinkering with the Schrödinger Equation unnecessary, i.e., it makes non-linear amendments redundant and allows the continuation of the peaceful coexistence between quantum mechanics and special relativity [26].

In the following, this proposal shall be somewhat detailed and some explications of how this might offer interesting perspectives are sketched, including some relationships to selected experiments, extant approaches and interpretations.

## 3. Time Is Relational

The concept of time has a long and rich history in itself. In CM, time is Newtonian, absolute and existing from its own nature, linearly passing without relation to anything external, without reference to any change of matter. Later, with special and general relativity, the universality and absoluteness of time has been overthrown. Interestingly, also inside QM, Newtonian time is (mostly tacitly) presupposed as a fixed background causal structure, i.e., the standard Schrödinger Equation as well as the Heisenberg picture work with linear time. An approach to rectify this and to devise a true quantum clock (Page-Wootters mechanism) and similar constructions claim their success by yielding essential Newtonian behavior for subsystems while the global system is stationary [27,28,29]. There are limits to this. For bounded quantum systems, no good quantum clocks can be constructed: apparently suitable quantum observables, which monotonically increase with Newtonian time, have a non-vanishing probability of running backwards [30].

This and the following leave out the very beginning of time, i.e., the first split moments where it all began with The Big Bang. An embedding in overall boundary conditions including the start of the universe shall come as a later topic following the more modest and restricted aim of first outlining a link between ordinary human everyday reality and QM in the sense advocated by Anton Zeilinger [2,31].

Time has been thought of as similar to space and, later, it has also been declared as devoid of any independent existence without space [32,33,34]. The discussion of absolute Newtonian space has reached an early summit when Gottfried Wilhelm Leibniz pointed out that space is only meaningful in the form of relations between positions [35]. Such ideas have later been elaborated by Ernst Mach and Albert Einstein [36,37].

Quite the same argument can actually be raised with respect to time. Leibniz, and later, amongst others, Einstein, saw it as the order of successive phenomena (which need to be distinguishable from each other).

Time in fact is relational at a very basic level, even without special or general relativity, as time is always measured relating to some type of “enduring” reference.

No clock is a clock without some memory. There are minimum requirements on traces, i.e., distinguishable and telling memory records (snapshots lacking suitable meta-information, e.g., involving irreversible dissipation, do not reveal their order nor their spacing).

Taking a simple pendulum as an example, one needs to recall that (where) the mass has started to move in order to take this as a basis for monitoring any change or process. For durations spanning more than one full swing, additional information storage is required, e.g., a little friction (energy transferred to a heat reservoir and increasing entropy there) to distinguish one period from another [38]. When building clocks based on cycles where friction is pushed into the background, some type of incremental and irreversible counting mechanism is mandatory to tell moments (periods) in time apart and keep track of any flow of time.

For a classic pendulum, its dynamics are always well defined whether specifically looked at or not. Moreover, for isolated CM set-ups without observing the system, we can tell the position of the mass for any selected point in time, and, upon observation, we find just this one predicted value. Uncertainties are not at all a matter of principle at the same level as with the case of an isolated quantum system.

Not only classical mechanisms but also useful quantum clocks need to generate observable time marks; CM “tick registers” are required for keeping an irreversible record of the clock time [29,39,40].

With full identity or full reversibility of repetitive states, no elapsing of time nor any direction of it can be told. Thus, even before many specific and rather involved problems with time can be identified, the notion appears a little more intricate at its conceptual foundation than is commonly thought [33,41,42,43].

Overall irreversibility is described by the statistical Second Law of Thermodynamics, which states that the total entropy of an isolated (sufficiently large) system can never decrease over time; only in cases where all processes are reversible does it stay constant. As an example, large increases in coherence times for the dephasing of a qubit are observed as a cavity is decoupled from its environment [44]. In small subsystems, where statistical unlikelihood poses not the same stringent constraint as for big systems, entropy (time) may fluctuate [45,46,47]. Still, isolated (quantum) systems spontaneously evolve rapidly towards the state with maximum entropy, i.e., thermodynamic equilibrium, and then stay most of the time there [48,49].

There is a well-defined condition for connecting activity (changes in time) with changes in information. That “Information is Physical” has been put on a solid theoretical basis by Rolf Landauer already 50 years ago [50,51]. Landauer’s principle states that irreversibly erasing one bit of information means increasing entropy by at least k*ln2; according to that principle, irreversible erasure in finite time goes with an unavoidable minimum cost in energy, for one bit [50]:ΔE ≥ kT ln2

This energy must be dumped to the environment, i.e., transferred to a heat reservoir. Landauer’s principle has been found generally valid in a large number of experiments, actually comprising CM and specific QM settings [52,53]. For quantum systems, basically the same principle holds as for classical systems, and instead of energy, other conserved quantities can be utilized [54,55]. It is the sum of the work required for measurement and erasure, which is principally bounded [56]. The limit will not be reached in general as it applies for quasistatic conditions; additional information in terms of coherence/entanglement can result in fluctuations and considerably higher dissipation upon erasure [57,58].

For QM, this might offer a way to pin down the shifting split, which is not even new: it is the merging in the flow of control, a two(or more)-to-one mapping of states, which is decisive [50,54,59]. Landauer’s principle can be derived from statistical mechanics with uncontroversial assumptions, and it is valid for non-equilibrium dynamics [59,60].

Landauer’s principle also dovetails with the Holevo bound stating that for n qubits only n bits of classic information are retrievable [61,62].

John Cramer reports Erwin Schrödinger saying that a state vector would collapse as soon as some macroscopic record of the result of a measurement is made, and Werner Heisenberg suggested a collapse occurs when a quantum measurement would pass from the domain of reversible processes to the domain of thermodynamic irreversibility [63].

With the uncontrolled transfer of a minimum amount of energy (or of another conserved quantity), unitary evolution of a quantum system, as described by the linear Schrödinger Equation, is interrupted [64]. Increasing the number of scattering events, coherence is increasingly lost [65]. Classical mechanics starts to reign, and the Second Law of Thermodynamics fully applies. For a specific case, the involved steps measuring an arriving particle have been described in great detail, highlighting in particular energy transfer [66]. Minute momentum transfers have been claimed to prohibit microscopic reversibility and thus produce the “origin of time” [67].

Before an interruption like this occurs, decoherence can be effective [68].

Von Neumann entropy grows with increasing decoherence in quantum systems, evolving with scattering processes [11]. Lesovik et al. claim that the overwhelming complexity of preparing time-reversed entangled quantum states can be seen as lying at the origin of irreversibility [11,12].

A full projective measurement in a quantum system irreversibly reduces the entire set of possible outcomes to a single specific one. This precludes complete reversal and thus means some loss of information and an increasing total entropy. Tradeoffs between information gain and the disturbance of a quantum state are close to the heart of QM [69]. True collapse as assumed by the standard Copenhagen interpretation, an event occurring, is here claimed be mainly accompanied by a minimum uncontrolled transfer of energy as given by Landauer’s limit.

Assigning a maximum amount of information of one bit to a basic quantum system (possibly even consisting of several entangled entities) fits the picture [31,61]. Reversible entanglement with an environment finds its end with an uncontrolled energy transfer effecting one of the fully entangled partners (and only to some degree in a weak measurement).

Conversely, in the absence of disturbance, entanglement can be swapped and new partners to a quantum object are recruited.

Avoiding uncontrolled entropy production, a prepared central quantum state can be recovered from an ancilla or the environment in a second step if they were sufficiently entangled, even if that original state is destroyed specifically between the start and end measurements [25,70].

With tightly engineered and controlled dissipation, an approximate Bell state in qubits can be produced deterministically and stabilized [71,72]. Preventing uncontrolled energy dissipation and harnessing pure dephasing-decoherence, electron spin correlations can be coherently transferred [73].

## 4. Fundamental Directionality

The collapse of the wave function according to the Copenhagen interpretation can be understood as asymmetric and as defining a clear arrow of time, each time a projective measurement is made. It points from QM → CM, and it has the direction of the general thermodynamic arrow of time in full accordance with Landauer’s principle [19,66,74,75,76].

The above-identified irreversibility applies to reconstructing the undisturbed quantum state, leading to a specific outcome, but in particular also to the earlier preparation stage, when and where that quantum system started out [46]. At this other end of the existence of some undisturbed quantum system, careful state preparation is required for a well-defined start which can, with the exception of The Big Bang, only be conceived of as belonging to CM, emphasizing an operational point of view [71,77,78,79]. In an experiment, e.g., the time-step goes from classical to quantum physics. The direction of time at that interface CM → QM is the thermodynamic one, and no problem with time has been seen here so far, as the Newtonian time in a laboratory seamlessly matches with the Newtonian time taken as a basis for unitary evolution expressed in a linear and deterministic Schrödinger Equation, the solutions of which are invariant under time reversal. This asks for some second thought, as deterministic details from the wave propagation are reduced to individual random detection events following the Born rule in the end.

Just the same as measurement, preparing a quantum state, reduces many options to one, both can be understood as erasing information concerning an earlier state, which is not fully accessible and cannot be reconstructed thereafter.

State preparation is a measurement, and it generally involves energy transfer to, and associated entropy increase in, the environment containing and constraining the quantum system, in which the special delicate state is realized [77,78,79,80]. At both (timewise) endpoints of a quantum mechanical system, thus some permanent traces are produced in CM, and none in between, i.e., inside an isolated quantum system, which obeys a unitary Schrödinger Equation. These records of events at the boundaries in the environment comprise all, which is principally fully accessible. Already Niels Bohr claimed that nothing at the quantum scale is real before being measured, and also Archibald Wheeler emphasized: we can only compare records of the past with the present “the past is not really the past until it has been measured, the past has no meaning or existence unless it exists as a record in the present” [81,82].

Contrasting the CM state immediately before the preparation of a quantum system with the CM situation after the measurement, many options have been reduced to one; certainly, some information has also been obtained. A many-to-one mapping actually can be claimed to have happened at both borders of the embedded quantum system in a CM frame. These local increases in order have a price in the form of increasing entropy outside that limited subsystem and outside the time interval between preparation and measurement. The arrow of time thus nicely points from the preparation to the measurement of a quantum state, both in the outside enshrining CM; it effectively bypasses or bridges the quantum domain.

This does not entail any circularity as a potential problem, which Tejinder Singh has warned of [83]. On the contrary, relatively sharp transitions (in time, but still non-instantaneous) between QM and CM break any ill-defined circular dependence, even if CM is accepted as the limiting case of QM. Along the same lines, decoherence accounts are released from the “blemish” of naively presupposing semi-classical time [5,6].

Classical states enclose the thus embedded QM realm at both ends, fixed and marked by enduring records, at least since time started quickly after The Big Bang with the universe in a very special, highly ordered, low-entropy initial condition. Such records need not be clear-cut snapshots of a situation possibly containing a detailed logging of earlier events; increases in overall entropy as in the case of a damped pendulum also qualify. It is reassuring also that Murray Gell-Mann and James Hartle, in their attempt to derive classical behavior from the sophisticated application or quantum mechanics and coarse graining, employ records but conclude that “generalized records of histories” need “not represent records in the usual sense of being constructed from quasi-classical variables accessible to us” [3].

Even non-measurements, i.e., (quantum) measurements completely without disturbing the measured object, as first envisioned by Mauritius Renninger, can be understood as relying on, in this case well-defined, CM framing [84].

Interaction-free measurements in a well-controlled set-up, with an object including knowledge about its prior location, can yield information about that object without changing its momentum, but then the momentum of the detection device will necessarily be altered upon absorption of the involved photon, as in the example by Lev Vaidman [85].

Quite generally, no result is a result without knowing what the measurement set-up was. Niels Bohr formulated a contextuality requirement: “the unambiguous account of proper quantum phenomena must, in principle, include a description of all relevant features of experimental arrangement” [86]. Archibald Wheeler spoke of the “great smoky dragon” with only its tail and head sharp [87].

This complementary embedding and framing of QM in a context of CM is claimed as necessary and marking limits and constraints for all meaningful interpretations of QM. At the same time, it preserves some overall (not tightly local) “distributed” locality as well as real causality and guarantees that no information or energy is transmitted between the endpoints with a velocity faster than the speed of light [88]. It has been shown that no-signaling in time, i.e., suitable statistical noninvasive measurability, is not only necessary but also sufficient for macroscopic realism [89].

The proposal here then is that in-between state preparation and measurement, in undisturbed QM, a concept of time, different from our standard one, applies. In the absence of irreversibility and without permanent records, time just is not “real” inside an isolated quantum system. Restricting the notion of time to its classical manifestation in CM (records), “timelessness” could be a label for the quantum world, QM. Lacking time as a fundamental pillar, QM objects, no matter whether waves, particles or their paths might be considered “unreal” [78,79].

“Timelessness”, as such, is not a new idea. It was proposed in a specific form by Albert Einstein. In a block universe, all of space and time is claimed to exist eternally [42,75,90].

Long known general findings fit nicely: the entropy in an isolated quantum system is constant, and that time is not a simple quantum observable has been pointed out already by Wolfgang Pauli [91]. Time cannot be described by a Hermitian operator; for an alternative view under specific conditions, see, e.g., [92,93].

In order to obtain real eigenvalues of a Hamiltonian, it is not really mandatory to have only Hermitian operators; the weaker condition of PT symmetry is sufficient [94,95]. Without unitarity of the time evolution, but with a balance of (energy-) loss and gain, stable states are possible whose overall probability does not change over time [96]. This can be linked in a unifying framework to the well-established quantum Zeno effect and the anti-Zeno effect [97].

Systems, which are characterized by time-dependent Hamiltonians and where energy is not conserved, clearly cannot be considered “isolated”. They are tied to boundary conditions and to laboratory time (CM) and are by definition, easy to accommodate in the Heisenberg picture of QM. With discrete time, “individual instants” might be considered timeless, and Born’s rule applies for measurements at those points. In turn, from the discreteness of time with a finite step size in the presence of a location-dependent force, microscopic entropy and time irreversibility can be derived via ensemble averaging [98].

A recent proposal for an evolving block universe acknowledges the omnipresent daily use of classical boundary conditions; the past is fixed and the future is (described as) open; the future simply does not exist at any experienced point in time yet (and thus, photons cannot emanate from there) [75]. Spacetime “growing” into the future as events unfold is a similar proposal by Avshalom Elitzur and Shahar Dolev [99].

## 5. Timelessness Strictly inside QM, Experiments with Slits

According to the famous statement by Richard Feynman, Thomas Young’s double slit experiment contains the essence of QM. Therefore, it seems only natural trying to look at how a fresh proposal of CM-constrained QM-timelessness might shed a little light on the paradigmatic double slit experiment in some of its different versions.

Packets of energy, photons or particles arrive in single locations as particles while exhibiting wave-like interference on a screen, which is placed at the opposite side of the source, far behind the slits (Fraunhofer regime). The interference pattern builds up when probes are sent one by one, and what is observed is self-interference for each single particle [100]. The same has, for example, also been concluded from experiments employing rather massive Fullerene buckyballs in the form of C_60_ and C_82_ [101,102].

Richard Feynman’s path integral approach does take into account all possible routes of a probe between source and detector, not only classical trajectories where action is stationary (at a minimum), but also non-classical paths [90].

Only recently has it been confirmed by extensive simulations that in a triple-slit set-up, simply assuming an interference between signals from slits, open one at a time, is not fully correct [103,104]. All paths in superposition allowed by the prevalent boundary conditions have to be taken into account; in particular including non-classical ones, i.e., looped trajectories.

In the case of three slits, experiments have actually shown that all conceivable paths contribute, and Sorkin parameters quantifying this have actually been measured [105,106]. Deviations from the superposition principle are not attributed to a violation of Born’s rule, but rather to an exquisite sensitivity to boundary conditions. Enhancing electromagnetic near-fields close to the slits by the excitation of surface plasmons accordingly strongly increases the contributions from non-classical paths [107].

Deliberately constraining and excluding looped trajectories by adding carefully placed absorbers allows for quantitatively controlling the magnitude of the Sorkin parameters, which depend also on the Gouy phase [106,108].

Timelessness in this context would simply mean that a particle or photon can in a sense effectively explore all of the possibilities permitted by the boundary conditions. With no time passing, there is also no shortage of time, and all possible transits do contribute according to their weight (based on the integrated action). Looking at the outside CM world, the occurrence of non-classic paths can be detected as a phase shift in the recorded interference pattern [108].

Complementarity as proclaimed by Niels Bohr implies that in a single experiment a quantum object either shows wave or particle characteristics. In particular, in the double slit experiment, there is no way of determining with certainty which path a particle took while still observing a wave-like interference pattern at a screen some distance behind the slits. The important point to observe now appears to be that this information (i.e., a record) certainly belongs to the CM world outside of the investigated quantum system. Whenever complete which-path information is available in principle, classical behavior is observed and interference is destroyed. This has been found in uncountable experiments for waves and particles including heavy-weights like Fullerenes [101,102]. For these it has been shown that emitting enough short-wavelength thermal photons, which would allow path determination, suppresses the interference pattern, and the same for collisions with gas molecules [109,110]. It is interesting to note that the uncontrolled energy transfer to the environment in these experiments is not too different from the corresponding Landauer limit.

One line of argumentation does not necessarily in all cases obviate or invalidate a second one; on the contrary, several somewhat independent lines of argument and derivation—the more diverse (but, of course, converging), the better—substantiate any result; e.g., wave-particle duality relations have been shown to be equivalent to entropic uncertainty relations [111].

Looking at the primary examples, the full determination of which way a particle travelled seems to imply the blocking or full detection of a particle, and thus some non-negligible energy transfer before, or rather instead of, the probe hitting the final screen. In light of the above, this would just mean that a collapse has occurred and QM has been left at that first occasion.

Particles and wave packets, which are strongly marked in order to disclose a path, e.g., by spin, show no (self-) interference pattern.

For cases with reduced information on the path, a deterioration of the interference fringes has been calculated and observed for ensemble averages; non-demolition experiments allow to explore the trade-off between particle and wave signatures, necessarily employing ensembles of probes, not specific single ones [112]. In experiments with two fully entangled probes, measuring one of the partners collapses the common state and thus collapses also the wave function of the partner [113].

If disturbance is kept to a minimum using weak measurements, it is possible to observe post-selected average trajectories of single photons in a two-slit interferometer [114]. Which-way experiments with a monitored momentum change along Bohmian trajectories yield a quantitative relation between the loss of visibility and the momentum disturbance accumulated during the propagation of the photons [115]. With entangled quantum particles in a related experimental set-up, non-locality yields “surreal trajectories” [116].

Similar to tests of Bell’s inequality, in delayed choice experiments as conceived of by Archibald Wheeler, any change is effected after the probe has left the source, but definitively before it is recorded [88,117]. Experiments have confirmed QM predictions employing single photons and particles, also with light from far distant quasars for random number generation closing locality loopholes [118,119,120].

With timelessness for the free particle/wave in between, it does not know when something happened and there is no well-defined moment in (nor record of) the pure QM phase, while boundary conditions are effective throughout. “Erasing” afterwards means filtering applicable sub-ensembles (classically recorded), which reestablishes an interference, strictly limited to the selected/filtered ensembles. Any actual choice brings suitable coincidences to the foreground and out of the hiding among the total observed events by means of considering only relevant corresponding events (records thereof).

Timelessness entails non-locality, a particle and its entangled partner in a sense are everywhere, continuously, always together (if sufficiently undisturbed). Provoking an event, i.e., interrupting anywhere/anytime, leads to one integral result following from the overall probability distribution and the intimate link between fully entangled partners.

In a variant of the double slit experiment, Shahriar Afshar has devised a layout, which seems to demonstrate a violation of the complementarity principle, as it is (incorrectly) understood as posited by the Copenhagen interpretation [121]. A grid is placed in between the detectors and the pinholes (serving as “slits” in this set-up). Particle and wave-like characteristics occur in one and the same set-up for single photons, and it seems possible to determine their path and observing interference fringes when accumulated while not perturbing with a measurement.

These findings have been taken as an argument for the transactional interpretation of QM, which proposes some hand-shake between retarded and advanced waves and thus describes atemporal pseudo-time conditions between emitter and absorber [63]. John Cramer, following the work of Wheeler, Feynman, Dirac and others, observes that the square of the wave function, which is decisive according to the Born rule, is the product of the advanced and retarded wave, which can be seen as kind of echo travelling back in time. The operation of complex conjugation is Wigner’s (“irreal”) time-reversal operator. With no real time passing for the isolated and free particles in between sender and receiver, there is plenty of opportunity to take the full boundary conditions into account, in particular, for the emission as well as absorption; contributions of forward and backward waves are thus included. Collapse, actually, occurs only at the detectors at the end.

The realization of a modified Afshar experiment using a Fresnel biprism and single photons yielded results fully compatible with the standard interpretation of QM including the complementarity relation [122].

Answering an objection that the measurements would not be simultaneous, Afshar claims that complementarity is violated as both findings would refer back to what “takes place” at the pinholes when a photon passes that plane.

An interpretation emphasizing the uncontrolled transfer of energy (collapse) could elucidate Afshar’s findings by the fact that the grid (the obstacle) is placed in areas which are not crossed by the main contributing “paths”, and at these locations, no interference pattern is effectively recorded. An interference pattern is just inferred; hardly any collapse takes place there and no time-stamp clearly visible in the statistics is generated.

Timelessness with the full probing of all possible options for an undisturbed wave function might offer a heuristic connection to the randomness intrinsic to QM. Chance results are what happens when from a wide range of different options, one is selected like in a lottery. In the classical world, statistics on the results for ergodic processes can be compiled by either repeating the drawing exercise over and over again at different points in time or at one time in parallel from many instantiations of that distribution, i.e., identical ensembles. Aside from the principal problems of having a quantum state over time resembling a composite quantum state at a single time, both appear hard, with only one particle on its way between source and detector [123].

Dropping the requirement of Hermiticity (in particular, conceding that time is not an observable) while keeping four other reasonable assumptions for a quantum state over time, allows treating quantum systems over time in the same manner as composite systems at a single time [123].

With all (timeless) potential outcomes available together, a single particle (or the universe during the The Big Bang) might constitute something like its own ensemble (with all possible superpositions), suggesting a link to a frequentist interpretation of probability.

With weak measurements, it is possible to characterize quantum trajectories, which bear some similarity to classical stochastic trajectories of particles interacting with a thermal reservoir [107,114,116]. In an open quantum system, it is even possible to derive from measured records a statistical arrow of time in measurement dynamics consistent with the macroscopic one when comparing probability densities of forward trajectories with time-reversed ones [46]. Without contact to a heat bath, the arrow of time is also constrained, analogous to the case with contact; investigating fluctuation theorems, it is the measurement-induced wave-function collapse inherent to information acquisition which evokes irreversibility [46,47].

The Born rule can be seen as a particular way of counting configurations, emerging from the factorization property of records as defined by Henrique Gomes and a reduction to the purely classical density [90]. The difference between a classical stochastic process and a quantum one can be traced to different sum rules for probabilities, as explained by Rafael Sorkin, i.e., for the QM case complex amplitudes are applicable instead of direct probabilities in CM [105]. Universality of QM is not required to derive Born’s rule from the standard QM measurement postulates with the very modest and reasonable assumption that choices in the description do not effect predictions [124].

Born’s rule has also been shown to result from picking outcomes with threshold detectors in a natural way from classical random signals for ergodic processes [125,126]. Some measure of stochastic ingredient is indispensable to arrive at Born’s rule; this might even be the unknown and fleeting exact timing since the beginning of the universe or some random background field [127,128].

Taking collapse seriously with durable records laid down only then and defining these specific moments, entails monotonic incremental time in distinguishable steps as well as causality, and it obviates back-acting handshakes with the future [23,63].

The proposal here to “understand” the behavior of probes in the double slit experiment then is that classical records are essential and only they constitute “real” time or time compatible with special and general relativity (first, for cases sufficiently far from extreme conditions at The Big Bang or a possible end of times in a very far future). Classical marking points mandatorily enclose every isolated (or very tightly controlled) quantum system, for which (internally) no “real” time passes in between state preparation and projective measurement [87].

Records of different types have been postulated before [3,34,38,75,90,129,130,131,132]. Mandatorily, it is distinguishable records which can be ordered according to their occurrence in succession (where available). Only records enable and define time. The term has especially been proposed for items which contain a whole history of events (“time capsules”), which had happened in mutually consistent histories at earlier times [34]. The existence of redundant records has been found to be a sufficient condition for redundant consistency in an attempt to explain an objective past from decoherence, describing sequences of events which take/took place in a closed quantum system [131,132]. This is effectively the case for all records marking events as here described. Records are linked and organized in non-strict hierarchies corresponding to the past light cones of particular events.

## 6. Tunneling

A next grade of “timelessness” could be seen in the measurements of tunneling times. QM allows particles to transcend barriers, which are too high to overcome according to the laws in CM; i.e., particles/waves tunnel through them. Inherently in QM, when there is a finite probability density attached to one side of the barrier, there is some at the other side, too. Since the discovery of QM, the question has been raised as to what time the tunneling process would take, or whether it would take any time at all, and whether any delay might be understood as a transit time [92,93,133]. Introducing dissipation, the predicted as well as the observed group delay increases linearly with the barrier length as expected for classical propagation [134,135]. The consensus in attoclock experiments now appears to be that no real time duration can be assigned to the very tunneling [136,137]. This means “timelessness” of the strictly quantum effect of “free” tunneling, zero time passing. Ossama Kullie offers an alternative view but it might be somewhat doubted as it is based on a specific time—an energy uncertainty relation. In fact, his account might still be consistent with the suggestions made here by considering properly the three separate phases of preparation, tunneling, and measurement [92,93,138,139]. The collapse of the wavefunction effected by the measurement, and, somewhat symmetrically, preparation, certainly are not instantaneous but are associated with some finite duration. For Larmor precession, taking into account the uncertainty relation for spin components is strictly required, and the associated time depends on the height and width of the barrier; the tunneling particle is tied (with only a little disturbance) to the outside CM world, i.e., via the magnet field in a weak measurement with man repetitions [140,141].

Just the same in each instantiation, state preparation and measurement of a tunneled particle would still need some time. Assuming an uncertainty relationship between energy and time, which is far from clear itself, it can be taken that with both of these steps, some short duration would be associated [77,142].

Even with zero time spent on tunneling itself, there is no problem with superluminal information transmission, as the probability of any useful signaling is negligible low [143].

Time–energy uncertainty relations have been discussed already by the founding fathers of quantum physics and ever since. The only thing clear so far is that time–energy uncertainty is not at the same level as others, e.g., position–momentum, and there are many facets to the topic [92,93,122,138,139,142,144]. The very basis of the difficulties with any time–energy uncertainty relation might actually consist of real time being only defined at/by durable CM recording points. This does not directly run counter to observations and applications of time–energy entanglement with (emission-)time and energy taken as continuous variables [145]. Just the same as with other pairs like polarizations in space, the entangled entities are only prepared as well as measured as (ultimately discrete) values outside the CM world.

The situation becomes especially intricate in cases where energy below some (Landauer) threshold is exchanged with a quantum system and/or, in particular, when this is tightly controlled to carefully minimize disturbing back-action from the environment [146]. “Collapse” need not be complete, not-involved superpositions can survive, and it is not instantaneous; smooth transits have been measured in experimental set-ups implementing undisturbed conditions close to ideal [147,148,149]. The experimental finding of the evolution of each completed jump taking time and being continuous, coherent, deterministic and controllable might be taken as implying proper time inside a quantum system, contrary to the “timelessness” advocated here. A second look reveals that time in this experiment was not established during the monitoring of the transition directly; not one clear-cut event or record has been created. The evolution was observed by looking at an auxiliary “bright” state and recorded outside in CM; only when knowing all limited options can the absence of an event at a sufficiently well-constrained time yield the same information as its occurrence [84]. The quantum system inside has no durable memory of which (“dark”) state it is or has been in. Differences in time scales allow combining the randomness and discreteness of individual jumps (on a long time scale) with a coherent and continuous evolution of such jumps (over a short time interval). Assuming some type of energy–time uncertainty relation, the latter has to be expected whatever the exact details of that relation are [139,142].

Avoiding “instantaneousness” renders a “collapse” more “physical” and “real”. Real effective “moments” are “thick” and have a minimum duration. At the same time, there is less need to push “collapse” and the generation of records to the epistemic and mathematical realm like in interpretations of QM as purely individual Bayesian updating.

Records cannot be arbitrarily sharp, not in CM and even less so in QM, where uncertainty relations prevail.

An important class of time intervals is commonly derived for the durations it takes for any reversible quantum process to unwind completely, for a (QM) system to return to its initial state. This would mark the other end of the scale for time, as there are good arguments that this, in interesting cases, could take forever [48,49]. Ideal projective quantum measurements in a finite temperature environment, which are faithful, unbiased and non-invasive, have been found to demand infinite resources [150]; even before, the identification of a system has been shown to be at best approximate [151].

Similar considerations actually are not confined to QM as full and exact measurable reversibility does not exist in any realistic CM system either. A common feature is that Poincare recurrence assumes infinitely accurately determined (and known) initial conditions, which have been debunked as un-physical [66,152]. As an example, most recent work shows that for three black holes orbiting each other, there is a fraction of constellations which for time-symmetric unwinding would demand local precision smaller than the Planck length [153].

Long before close to an end of time, another limit is encountered. Even with arbitrarily accurate starting conditions, a “predictability horizon” in many cases limits time intervals for which interesting and meaningful forecasts are possible, i.e., the range for which developments and approximate expectation values for acceptable errors can be given (Lyapunov time).

Principal limits to the achievable accuracy of boundary conditions and measurements (and subsequently, predictions), both in QM and CM, could be seen as somewhat blurring the distinction between these respective realms. On a conceptual level, it is obvious that with two areas closely bordering each other and actually interleaved with each other, with one of them comprising intrinsic uncertainty, the second one cannot also exhibit infinitely sharp conditions (when the step in between cannot unphysically be infinitely sharp).

An intimate link relating distinctions in time to causality and to entropy is obvious; it can convincingly be argued that time has to come in discrete increments for any meaningful concept of causality [98,154].

Stretching the frame here to its maximum, situations with effectively “infinitely long” as well as with “zero” time passing might be labeled as “timeless”. What remains in a limited middle ground would then be just “permanent” traces, entropy increased in the environment, i.e., records in their relative (causal) ordering. It matches nicely that in the quasi-static limit, any logically irreversible computation can be performed in a thermodynamically reversible manner. Only if erasure is performed with a finite velocity the erasure becomes thermodynamically irreversible [50,56].

Real causality is inseparably linked to processes unfolding over time and total entropy accumulation. Causality does not go backwards, “events do not unhappen” is the formula coined by Lee Smolin [155]. With an identified order of classical records, their potential causal dependences are constrained; (the record of) an effect can never precede (the record of) its cause. This cannot simply/fully be transferred to inside QM, which allows more complex and entangled connections, alas, without producing durable records inside [122,156].

Taking into account that it is only classic records, which at the end can be unambiguously ordered in one or more light cones, findings where the future seems to influence or determine the past thus do not disturb everyday reality. This situation can only be (weakly) observed in sophisticated (QM) experiments and can possibly be harnessed in quantum computers [118,119,120,157,158]. In an interferometer experiment, the succession of states and their influence has been brought into superposition. Two stages/operations (not “events” involving energy exchange and leaving records) could not be ordered according to any causal relation in coincidence measurements after photons have all passed through the set-up [158]. While correlations, which could not be understood in terms of any definite order, have also been found with a task involving communication between two local partners in a framework strictly inside QM and without any global causal structure, in a classical limit, global causal order always arises [159].

Starting from small isolated quantum systems, a hierarchy of scales can be built including the meshing of relevant time scales [160,161]. “More is different” has been proclaimed by Paul Anderson and before him by Nicolai Hartmann and Hermann Haken [162,163,164]. Contrary to allegations by trivial accounts of reductionism, “timelessness”, at a most fundamental level, need not conflict with time emerging at higher levels.

## 7. The Mosaic

Taken together, contained “timelessness” might be a suitable label, which allows some “comprehension” by applying a sensible name for the peculiar conditions of isolated (very tightly controlled) systems in the underlying quantum world, which appear very strange and are distinctly different from what we know in the one world of classical physics, in which humans bodily live. Staying consistent with observations, with the well-established QM formalism, and also with some basic common-sense plausibility seems possible on a coarse and still meaningful level.

Amending the Copenhagen interpretation rudimentarily with (at last) a quantitative physical correlate for “collapse” given by the Landauer Principle and thus anchoring QM systems in the reality of CM, the importance of durable records is emphasized. At the same time, the role of conscious observers is downgraded to being mostly inconsequential. Thus embedded in CM and the general flow of time as witnessed by increasing overall entropy, quantum states are real to a certain extent. Isolated quantum systems can be seen as “timeless”, while still constituting “objective elements of reality”. Inside QM, neither records nor the order of stages are determined; there is neither causality, nor retro-causality, nor both (at the same no-time). It might be seen as paradox that “timelessness”, when strictly contained in QM by classic framing in the real world, animates a “timeless” static block universe, which knows only unitary quantum physics, to be real, evolving, and open. Concerning questions relating to circularity, the speed limit for light fully applies in the accessible CM world, and a second mechanism also makes sure that effective “leaps” and “jumps” cannot be truly instantaneous: some time–energy uncertainty relations certainly prevent unphysical infinitely fast changes.

A little light is shed even on subjective interpretations where consciousness interaction is claimed to be responsible for something like a collapse in a wave function; there might be cases when this is the first contact to CM, which triggers irreversible records, in particular, in sophisticated thought experiments.

In summary, it is claimed that the proposal here obviates all types of “queer” interpretations and allows for a “comprehensible” link between our everyday classical environment and bizarre quantum systems; at the same time, the well-proven mathematical formalism stays practically the same and untouched.

All real clocks are fundamentally thermodynamic. A most recent result shows that timekeeping has associated costs; a fundamental universal relationship between entropy production and clock accuracy applies in both quantum and classical regimes [165].

Timelessness strictly inside the quantum realm with quantum systems embedded in the complementary classical world leaves an open end at the very beginning of the universe. Widely accepted notions of inflation might be rendered obsolete there. It goes far beyond what is possible in this short paper, but it could be well worth while attempting to correct the applicable time scale instead of finetuning an inflationary potential and process. In light of the above, the concept of (classical, incremental) time simply would not fully apply before first structures, i.e., durable records, can form, conserved in a way (through their impact) and effective later (observable) in our epoch.

The working hypothesis for the peculiar boundary conditions at The Big Bang: no space, no records, no time, no entropy.

Claus Kiefer coarsely describes the emergence of space and time from the fundamental timelessness of quantum gravity, which appears to fit with the account advocated here [166]. Entropy production only starts with an increasing scale factor. Reversing the usual logic of argumentation, Erik Verlinde proposes gravity in turn to emerge as an entropic force once space and time themselves have emerged [167].

Modular spacetime, with relative locality, offers another promising alternative, while apparently not necessarily contradicting, approach [168].

Notably, a “slow beginning” of time can be interpreted as an effectively changing (initially much higher) speed of light. Swapping decoherence for collapses and entropy-production with records, classical irreversibility is obtained, while leaving the arguments relating to the beginnings (The Big Bang without inflation) pretty much the same, and averting a Big Crunch.

In the end, the widest conceivable consistency is all that can meaningfully be asked for.

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
