# Peer review of "Timelessness Strictly inside the Quantum Realm"

_entropy, 2021, doi:10.3390/e23060772_

Round 1

Reviewer 1 Report

See attached file.

Reviewer 2 Report

In my opinion, this work is an important contribution to the discussion of the basics of quantum theory and the problem of the role and emergence of time in the quantum and classical worlds.

I recommend publishing this work.

Reviewer 3 Report

This paper is interesting as such. I find it difficult to be read, because I prefer to read discussion based on equations. But, the conclusions of the author can be considered interesting, and shareable. It is difficult to suggest how to improve the paper because it is a continuous speech.

I suggest to introduce some more equations in relation to the discussion, with particular interest on the Landauer approach.

Moreover, a deep analysis of time has been developed by Kuzemsky et al. on Entropy in 2020. I suggest to consider it and to consider also some references quoted in it, also from an experimental point of view.

I suggest also to improve the discussion on Einstein, by summarising its approach also by introducing some analytical results.

Can the author quote and summarise also some experimental results?

Reviewer 4 Report

This paper is a kind of personal testimony of the author's view on
quantum realm and on time. It is hard to find any new result; this
is at best a new presentation of many views and previous results.
As such, this is not a standard research paper. However, I enjoyed
reading this paper. I am sure some readers of Entropy would also
enjoy it. Also, the more than 150 references will be useful to some
readers. In short, this is a pleasant well written broad and sometimes
deep presentation of the author's opinion on important and timely
aspects of today's physics.

Round 2

Reviewer 1 Report

See attached pdf-file

Reviewer 3 Report

The authors have addresses all my suggestions. I suggest to accept the paper.

Author Response

Dear unknown colleague,

thank you for your effort! best regards and wishes,

Knud Thomsen

Round 3
